# Patients with Primary Central Nervous System Lymphoma Not Eligible for Clinical Trials: Prognostic Factors, Treatment and Outcome

**DOI:** 10.3390/cancers13122934

**Published:** 2021-06-11

**Authors:** Sabine Seidel, Michelle Margold, Thomas Kowalski, Alexander Baraniskin, Roland Schroers, Agnieszka Korfel, Eckhard Thiel, Michael Weller, Peter Martus, Uwe Schlegel

**Affiliations:** 1Department of Neurology, University Hospital Knappschaftskrankenhaus, Ruhr University Bochum, In der Schornau 23-25, D-44892 Bochum, Germany; michelle.margold@kk-bochum.de (M.M.); thomas.kowalski@kk-bochum.de (T.K.); uwe.schlegel@kk-bochum.de (U.S.); 2Department of Hematology and Oncology, University Hospital Knappschaftskrankenhaus, Ruhr University Bochum, In der Schornau 23-25, D-44892 Bochum, Germany; alexander.baraniskin@ruhr-uni-bochum.de (A.B.); roland.schroers@kk-bochum.de (R.S.); 3Department of Hematology and Oncology, Charité Berlin, University of Berlin, Hindenburgdamm 30, D-12203 Berlin, Germany; korfel_agnieszka@lilly.com (A.K.); eckhard.thiel@charite.de (E.T.); 4Department of Neurology, University Hospital and University of Zurich, Frauenklinikstrasse 26, CH-8091 Zürich, Switzerland; michael.weller@USZ.CH; 5Department of Biostatistics and Clinical Epidemiology, University of Tübingen, Silcherstr. 5, D-72076 Tübingen, Germany; peter.martus@med.uni-tuebingen.de

**Keywords:** primary central nervous system lymphoma, polychemotherapy, high-dose methotrexate, cytarabine, Bonn protocol, non-study patients

## Abstract

**Simple Summary:**

Many patients with primary central nervous system lymphoma (PCNSL) participate in clinical trials. The inclusion criteria for these trials are largely uniform among various trials on first-line treatment. Therefore, there is a lack of data on therapeutic management and prognostic factors for patients not fulfilling these inclusion criteria. Here, we retrospectively analyzed treatment, outcome and prognostic factors of 34 patients of our center who did not fulfill inclusion criteria for clinical trials, and compared those results with data from the largest study of PCNSL patients, the G-PCNSL-SG-1 (German PCNSL Study Group 1) trial.

**Abstract:**

Patients with primary central nervous system lymphoma (PCNSL) not fulfilling inclusion criteria for clinical trials represent an underreported population. Thirty-four consecutive PCNSL patients seen at our center between 2005 and 2019 with exclusion criteria for therapeutic trials were analyzed (non-study patients) and compared with patients from the G-PCNSL-SG-1 (German PCNSL Study Group 1) study (study patients), the largest prospective multicenter trial on PCNSL, comprising 551 patients. Median follow up was 68 months (range 1–141) in non-study patients and 51 months (1–105) in study patients. Twenty-seven/34 (79.4%) non-study patients received high dose methotrexate (HDMTX), while seven/34 (20.6%) with a glomerular filtration rate (GFR) < 50 mL/min did not. Median overall survival (OS) was six months (95% confidence interval [CI] 0–21 months) in those 34 non-study patients. The 27 non-study patients treated with HDMTX were compared with 526/551 G-PCNSL-SG-1 study patients who had received HDMTX as well. Median OS was 20 months (95% CI 0–45)/21 months (95% CI 18–25) in 27 non-study/526 study patients (*p* = 0.766). Favorable prognostic factors in non-study patients were young age, application of HDMTX and early response on magnet resonance imaging (MRI). If HDMTX-based chemotherapy can be applied, long-term disease control is possible even in patients not qualifying for clinical trials. Initial response on early MRI might be useful for decision on treatment continuation.

## 1. Introduction

Primary central nervous system lymphoma (PCNSL) is a highly aggressive diffuse large B-cell lymphoma that accounts for 1.9% of primary brain tumors [1]. Treatment for PCNSL has improved significantly during the last two decades mainly due to the invention of high-dose methotrexate (HDMTX)-based conventional [2,3,4,5,6,7,8,9,10] or myeloablative [11,12,13,14,15] chemotherapy. Two-year overall survival (OS) rates of 32–87% within clinical trials have been achieved with those regimens.

Nevertheless, a fraction of patients does not meet inclusion criteria for clinical trials. For this patient group, no treatment standard exists, and prognosis is usually poor. Eastern cooperative oncology group (ECOG) and Karnofsky performance (KPS) status are commonly used to determine eligibility for clinical trials. Some trials use a general cut-off for exclusion with a KPS < 40 [9], ECOG < 4 [4,12,13] or ECOG < 3 [10]; others exclude patients with a KPS < 60 independent of PCNSL or KPS < 30 because of PCNSL [7]. In addition, comorbidities compromising the ability to give informed consent and potentially hampering compliance; e.g., severe psychiatric disorders, generally prevent participation in clinical trials [2,4,10,11,12,13,14]. While some reports suggest effectivity of polychemotherapy in post-transplant patients [16] and in patients with autoimmune diseases requiring medical immunosuppression [17], most clinical trials exclude immunosuppressed patients [3,6,7,9,10,12,13,14]. Other active malignancies or previous malignancies within five years prior to diagnosis of PCNSL also result in exclusion from clinical trials [2,3,6,7,9,10,12,13,14]. Adequate renal function is a prerequisite for the application of HDMTX, which is the backbone of efficient chemotherapy protocols in PCNSL. Patients with a glomerular filtration rate (GFR) < 50 mL/min [6,7,8,11] (in some trials < 60 mL/min [2,3,4,10,14]) are not eligible for HDMTX and are excluded from clinical trials, including HDMTX-based chemotherapy. While some clinical trials do not consider patients with an ejection fraction (EF) of < 45% [4] or < 50% [8,10,11] or heart insufficiency NYHA (New York Heart Association classification of heart disease) class III b or IV [2,3], others exclude patients with “severe heart diseases” [7,12,13].

In this retrospective series, we analyzed outcome and prognostic factors of 34 consecutive patients treated at our institution between 2005 and 2019 who did not meet inclusion criteria for clinical trials in PCNSL as described above. We further compared the outcome of those 27 patients from this cohort (“non-study” patients) having received HDMTX with outcome measures of 526 patients treated within the German PCNSL Study-Group 1 trial (G-PCNSL-SG-1, “study” patients), since these 526/551 had fulfilled inclusion criteria for the trial and had been treated with at least one cycle of HDMTX.

## 2. Materials and Methods

The non-study patient population comprised all patients with histologically confirmed PCNSL that had been referred to our hospital from January 2005 until December 2019 and did not fulfill inclusion criteria for therapeutic trials. A full-body FDG PET or CT of neck, chest and abdomen, and a bone marrow biopsy at staging, were performed in all patients to rule out systemic lymphoma. All patients analyzed showed one or more of the following exclusion criteria for clinical trials in PCNSL: KPS < 50 independent of diagnosis of PCNSL, KPS < 30 because of PCNSL, comorbidity preventing patients from giving informed consent, immunosuppression, previous malignancy within five years prior to diagnosis of PCNSL, heart insufficiency NYHA III b or IV, EF < 50%, or GFR < 50 mL/h.

All patients with a GFR ≥ 50 mL/min received HDMTX-based polychemotherapy. One of these patients received chemotherapy according to the original Bonn protocol [3], while all other patients received a modified Bonn protocol [18] (Table 1) after obtaining detailed information on patients and informing their caregivers on the associated risks. Chemotherapy included systemic HDMTX (Medac, Wedel, Germany) (3000–5000 mg/m^2^), ifosfamide (Stada, Bad Vilbel, Germany) (800 mg/m^2^) and cytarabine (Accord, Munich, Germany) (AraC, 3000 mg/m^2^). Treatment protocols changed in different time periods: Therefore, depending on the date of diagnosis, systemic rituximab (Hexal, Holzkirchen, Germany) (2009–2019), intracerebroventricular (ICV) treatment with methotrexate (Hexal, Holzkirchen, Germany), prednisolone (Merck, Darmstadt, Deutschland) and AraC (Stada, Bad Vilbel, Germany) (2010–2019 for patients < 65 years; 2015–2019 for all patients) or intrathecal (ITH) treatment with liposomal AraC (Mundipharma, Frankfurt, Germany) via lumbar puncture (2005–2009) were part of the chemotherapy applied. Patients with a GFR < 50 mL/min received individualized treatment (chemotherapy with rituximab and temozolomide (TMZ) (TMZ was prescribed for outpatient care without specification of the manufacturer) (one cycle = 28 days, rituximab 375 mg/m^2^ day 1 and 14, TMZ day 2–6 at standard dosage; see also Appendix A), whole brain radiotherapy (WBRT, 45 Gy) or palliative treatment).

First (“early”) MRI control was performed after two cycles (2005–2009) or after three cycles (2010–2019) of treatment. Response to therapy was assessed according to IPCG (International Cooperative PCNSL Group) criteria [19]. Follow-up was evaluated for the purpose of this study until June 2020 (censoring date 30 June 2020.). All patients were seen after completion of therapy by serial follow-up, including neurological and ophthalmological examination, as well as cerebral MRI (and cerebrospinal fluid [CSF] examination in case of suspicion of relapse) according to international guidelines [19]. Data on patient characteristics, clinical presentation, laboratory data, imaging data, treatment and current life situation were collected by chart review. No personal or telephone contact with patients or general practitioners was necessary. Written informed consent was not obtained for this retrospective analysis. The local ethics committee approved the study (registry number 20-6818-BR).

OS was calculated from the date of histologic diagnosis to death of any cause or last date of follow-up. Progression-free survival (PFS) was defined as the time from date of histologic diagnosis to progression, death of any cause (if progression was not determined) or last date of follow-up. OS and PFS were estimated by the Kaplan–Meier method. Log-rank tests were used to compare OS and PFS between groups.

To relate treatment results achievable in patients not qualifying for clinical trials to those of patients treated within a trial, we compared the present results with those of patients from the G-PCNSL-SG1 trial. Therefore, we compared the results from the non-study population confined to 27 patients having received HDMTX to 526 G-PCNSL-SG1 patients, since those patients had also received at least one cycle of HDMTX. Furthermore, we investigated the effect of adjusting the comparison for age and KPS. Response rates were compared using the chi-square test, and survival was analyzed using the Cox proportional hazard model including 95% confidence intervals [CIs] for the hazard ratio and Wald tests. The level of significance was 0.05 (two-sided). SPSS (Statistical Package for the Social Sciences) for Windows (release 26) was used for all statistical analyses.

## 3. Results

### 3.1. Patient Characteristics

Between January 2005 and December 2019, 208 patients with PCNSL were referred to our center. Of those, 30 patients were treated within a formal clinical trial, and 40 patients were included in a prospective registry. A total of 138 patients were not included in a trial because there was no ongoing trial at our center at the time of diagnosis. Those 138 would have been eligible for inclusion in a clinical trial. Thirty-four patients did not meet inclusion criteria for a clinical trial. These 34 non-study patients were included in this analysis. All patients had histological diagnosis of diffuse large B-cell lymphoma. One patient was lost to follow-up. This patient was not able to receive HDMTX because of impaired renal function. The median follow-up for surviving patients was 68 months (range 1–141 months). The median age was 74.5 years, and the median KPS was 40. For details on patient characteristics, see Table 2.

Reasons for exclusion from clinical trials in non-study patients were: KPS < 50 independent of diagnosis of PCNSL (*n* = 1), comorbidity influencing informed consent (*n* = 2), KPS < 50 plus comorbidity influencing informed consent (*n* = 4), KPS < 30 because of PCNSL (*n* = 5), KPS < 30 because of PCNSL plus heart insufficiency (*n* = 1), immunosuppression (*n* = 7; long-term corticosteroids in *n* = 4, subcutaneous methotrexate (MTX) in *n* = 2, subcutaneous MTX and etanercept in *n* = 1), previous malignancy within five years prior to diagnosis of PCNSL (*n* = 3; one breast cancer, one bladder cancer, one prostate cancer), heart insufficiency (*n* = 4), GFR < 50 mL/h (*n* = 5), GFR < 50 mL/h plus heart insufficiency (*n* = 1), or GFR < 50 mL/h plus immunosuppression (*n* = 1) (see also Appendix A).

### 3.2. Treatment, Response and Survival

Seven patients (20.6%) were not eligible for HDMTX-based therapy. Response after treatment was progressive disease (PD) during treatment with rituximab and temozolomide in three and partial remission (PR) after WBRT in one. Three patients refused further treatment.

All 27/34 (79.4%) patients with a GFR of ≥ 50 mL/min received HDMTX-based polychemotherapy as first-line treatment. Response in these was complete remission or complete remission unconfirmed (CR/CRu) in 17 patients (63.0%), PD in six patients (22.2%), treatment-related death in two patients (7.4%), or premature treatment termination because of poor clinical state in two patients (7.4%) after having achieved PR. For details on treatment characteristics in non-study patients, see also Appendix A.

Median overall survival (OS) was six months (95% CI 0–21 months), 3-year OS was 30.9% (13.9–47.9%) in 34 non-study patients. In those 27 having received HDMTX, it was 20 months (95% CI 0–45 months) with a 3-year OS 37.4% (18.6–56.2%).

### 3.3. Prognostic Factors in Non-Study Patients

In a univariate analysis, favorable prognostic factors were age, application of HDMTX, achievement of CR/CRu at first (early) MRI control and achievement of CR/CRu after completion of therapy. Median OS for patients < 65 years was not reached, while median OS for patients ≥ 65 years was two months (95% CI 0–5 months; *p* = 0.006). Median OS for patients who received HDMTX-based chemotherapy was 20 months (95% CI 0–45 months), for patients not eligible for HDMTX two months (no CI can be calculated [20], *p* = 0.003; Figure 1). Median OS for patients who had CR/CRu at first MRI control was 43 months (95% CI 10–24 months), while for patients without CR/Cru, it was 2 months (95% CI 1–3 months; *p* = 0.001; Figure 2). Median OS for patients who had CR/CRu after completion of treatment was 43 months (95% CI 0–94 months), while for patients without CR/Cru, it was 2 months (95% CI 1–3 months; *p* < 0.001). No significant difference in OS was found for sex, KPS, involvement of deep brain structures, elevation of lactate dehydrogenase, CSF protein elevation, anemia at diagnosis of PCNSL or application of rituximab.

### 3.4. Comparison of Non-Study Patients and Study Patients

Median follow-up in 526 study patients from the G-PCNSL-SG-1 trial was 51 months (range 1–105). Median age in study patients was 63 years, and median KPS was 70. In study patients, CR was observed in 182/526 (34.6 %) patients compared to 17/34 patients (63.0%).

In study patients, median OS was 21 months (95% CI 18–25 months) with a 3-year OS of 38.3%. In 27 non-study patients having received HDMTX, it was 20 months (95% CI 0–45 months) with a 3-year OS of 37.4% (*p* = 0.766, Figure 3).

For comparison between non-study and study patients, different models were calculated: (1) naïve comparison 27 vs. 526 and (2) comparison adjusted for KPS and age. The “naive” analysis of 27 non-study patients vs. the 526 patients of G-PCNSL-SG-1-study led to a hazard ratio of 0.93. After adjustment for age and KPS, the hazard ratio decreased to 0.48 (*p* = 0.005, Table 3).

## 4. Discussion

In this retrospective series, 34 patients not eligible for clinical trials in PCNSL were analyzed regarding treatment, outcome and prognostic factors. With the exception of seven patients with a GFR < 50 mL/min, all others received HDMTX-based polychemotherapy. Three-year OS was 37.4% in this latter cohort. Except for the subgroup of patients with severe renal insufficiency, long-term survival of more than three years was observed for a fraction of patients in all subgroups independent of the reason for exclusion from clinical trials. As in many clinical trials on PCNSL [3,6,7,21,22], age was an important favorable prognostic factor in this cohort. The application of HDMTX, a long-known prognostic factor in PCNSL treatment [23], also was prognostic in this series, despite the potentially higher risk of toxic side effects in this frail patient cohort. Further, an early complete response was associated with a significantly longer OS in our patient population. This has already been reported for patients treated within a clinical trial [24], but might be particularly useful for decisions on treatment continuation in these “high-risk“ patients.

Studies on outcome and prognostic factors in patients with PCNSL in a “real-life” setting have already been published; however, these studies either included many patients who would have been eligible for a clinical trial but did not participate in one [25], or included patients who had actually participated in a clinical trial [26,27]. One series focused explicitly on patients treated “outside clinical trials”; however, in this series, reasons for not being included in a clinical trial were lack of histological diagnosis and no ongoing trial in many patients [28]. In the present series, we only analyzed patients with at least one exclusion criteria for clinical trials in PCNSL.

Previous results of retrospective studies on intensive chemotherapy protocols for immunosuppressed patients had shown encouraging results [16,17], which was analogous to the outcome of immunocompromised patients in this series.

Our data further suggest that intensive chemotherapy is feasible in patients in a particularly compromised clinical state (KPS < 30) at first diagnosis or in the case of (mostly psychiatric) comorbidities influencing informed consent, as long as the general physical condition allows application of chemotherapy.

For most patients with malignancies diagnosed within five years prior to diagnosis of PCNSL and patients with severe heart insufficiency, who did not have an additional contraindication for HD-MTX, chemotherapy could be applied without major safety concerns and with good results in this frail cohort. It must be considered that these prior conditions themselves can limit survival. Since only three patients with prior malignancies and five patients with severe heart insufficiency were included in this study, larger studies are required to confirm our findings.

Only for the subgroup of patients with severe renal insufficiency (GFR < 50 mL/min), and therefore not able to tolerate HDMTX, effective treatment hardly exists and early initiation of palliative treatment should be considered, while treatment attempts with rituximab plus temozolomide or with WBRT seem justifiable in some.

The results of this series are limited by the fact that it was a retrospective monocentric analysis of a rather small number of included patients. The size of the patient population was limited here by the design of the study itself; a multicentric study focusing on this particular patient population was hardly feasible.

We aimed to compare treatment results of patients not eligible for a clinical trial with those having been achieved within a clinical study. We had the opportunity to carry out this analysis by comparison with the original data of the largest PCNSL trial [7]. Even though the difference in number of patients within the two groups was considerable, we analyzed the data of *n* = 526/551 patients from the G-PCNSL-SG-1 trial who had received at least one cycle of HDMTX. Those were compared with the 27 patients of our cohort, who had also received HDMTX. Outcome measures of the 27 non-study patients were comparable to those of 526 study patients from the G-PCNSL-SG-1 trial according to univariate analysis (hazard ratio 0.93); however, after adjustment for age and KPS, there was a significant difference in favor of the non-study patients (hazard ratio 0.48, *p* = 0.005). These results may have been influenced by two important factors. It must be considered that the multicenter design of the G-PCNSL-SG-1 trial by itself may represent a negative prognostic factor, since study patients within a multicenter trial often are treated in less-experienced centers, which has been shown to be of negative prognostic significance [29]. An additional unfavorable prognostic factor might be the time period in which patients had been treated, as outcome and survival of PCNSL patients has improved significantly over the last two decades [30]. While the non-study patients had been treated between 2005 and 2019, patients within the G-PCNSL-SG-1 trial had been recruited between 2000 and 2009 [7]. An analysis regarding the prognostic impact of era of treatment in our population was not possible from a statistical point of view, since only five patients in our study had been diagnosed between 2005–2009.

Nevertheless, in comparison to study patients, the results achievable in “high-risk” non-study patients were promising, and in our view justify the application of intensive treatment to patients not fulfilling inclusion criteria for clinical trials as long as treatment with HDMTX is possible.

## 5. Conclusions

In summary, we concluded that if patients are eligible for HDMTX-based chemotherapy, the possibility of long-term survival exists even in a patient population with poor clinical prognostic factors and not eligible for clinical trials. Initial response in MRI might be useful for guidance in therapeutic decisions. These results should prompt clinicians to treat “catastrophic” PCNSL patients with regimens evaluated within clinical trials as long as those are applicable.

## Figures and Tables

**Figure 1 cancers-13-02934-f001:**
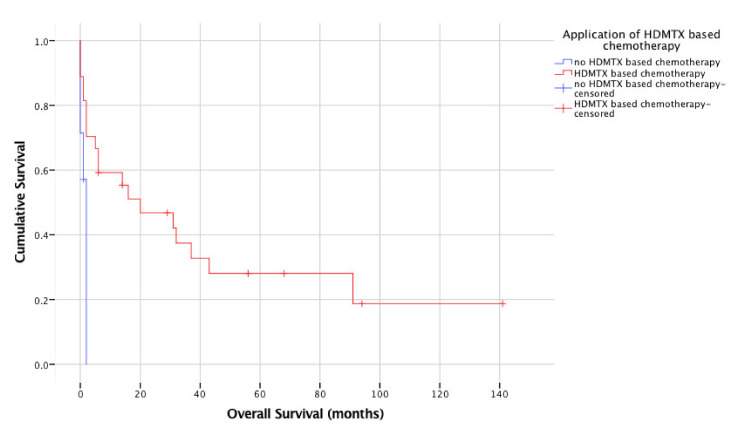
Overall survival according to application of high-dose methotrexate (HDMTX) (*n* = 34).

**Figure 2 cancers-13-02934-f002:**
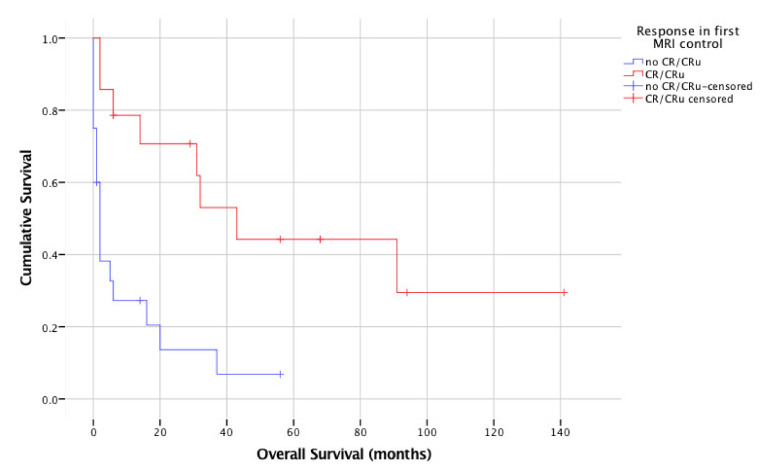
Overall survival according to response in first “early” MRI (*n* = 34).

**Figure 3 cancers-13-02934-f003:**
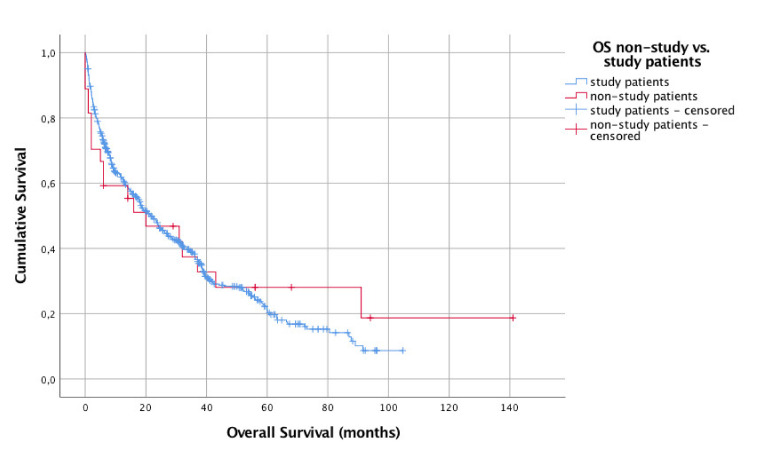
Overall survival of non-study (*n* = 27) vs. study patients (*n* = 526).

**Table 1 cancers-13-02934-t001:** Modified Bonn protocol.

**Chemotherapy**	**Day 0**	**Day 1**	**Day 2**	**Day 3**	**Day 4**	**Day 5**	**Day 6**
Cycles 1–3 (1 cycle = 2 weeks)	x	x					
Rituximab 375–500 mg/m^2^ IV^a^							
MTX 3000–5000 mg/m^2^ IV				x			
Ifosfamide 800 mg/m^2^ IV			x		x		
Liposomal AraC 50 mg ITH^b^			x				
		x					
Cycles 4 + 6 (1 cycle = 3 weeks)			x				
Cytarabine 3000 mg/m^2^ IV			x				
Liposomal AraC 50 mg ITH^b^							
MTX 2,5 mg + prednisolone 3 mg ICV^c^				x	x	x	
AraC 10 mg ICV^c^							x
		x					
Cycle 5 (1 cycle = 2 weeks)^d^							
MTX 3000–5000 mg/m^2^ IV			x	x			
Ifosfamide 800 mg/m^2^ IV			x		x		
Liposomal AraC 50 mg ITH ^b^			x	x			
MTX 2,5 mg + prednisolone 3 mg ICV^c^					x		
AraC 10 mg ICV^c^						x	

^a^ Patients treated 2009–2019; ^b^ patients treated 2005–2008; ^c^ patients < 65 years 2010–2019, patients ≥ 65 years 2015–2019; ^d^ cycle 5 was repeated for patients < 65 years. Abbreviations: AraC, cytarabine; ICV, intracerebroventricular; ITH, intrathecal; IV, intravenous, MTX, methotrexate.

**Table 2 cancers-13-02934-t002:** Patient characteristics.

Characteristics	All Non-Study Patients (*n* = 34)	Non-Study Patients with HDMTX (*n* = 27)	Study Patients in G-PCNSL-SG1 (*n* = 526)
Age, years			
Median (range*)	74.5 (65–77)	71 (62–76)	63 (55–69)
< 65 years	9 (26.5%)	9 (33.3%)	291 (55.3%)
≥ 65 years	25 (73.5%)	18 (66.7%)	235 (44.7%)
Sex			
Male	13 (38.2%)	12 (44.4%)	299 (56.8%)
Female	21 (61.8%)	15 (55.6%)	227 (43.2%)
KPS			
Median (range*)	40 (30–50)	40 (20–50)	70 (50–90)
Involvement of deep brain structures			
Yes	27 (79.4%)	22 (81.5%)	data not available
No	7 (20.6%)	5 (18.5%)	
Lactate dehydrogenase			
Normal	18 (52.9%)	12 (44.4%)	198 (37.6%)
Elevated	15 (44.1%)	14 (51.9%)	109 (20.7%)
Not done	1 (3%)	1 (3.7%)	30 (5.7%)
No specification	0	0	189 (35.9%)
CSF cytology			
Positive	2 (5.9%)	1 (3.7%)	44 (8.4 %)
Negative	18 (52.9%)	17 (63.0%)	294 (55.9%)
Suspect	0	0	23 (4.4%)
Not done/no specification	14 (41.2%)	9 (33.3%)	165 (31.4%)
CSF protein elevation			
Elevated	14 (41.2%)	13 (48.1%)	154 (29.3%)
Normal	6 (17.6%)	5 (18.5%)	150 (29.3%)
Not done/no specification	14 (41.2%)	9 (33.3%)	222 (42.2%)

Abbreviations: CSF, cerebrospinal fluid; DLBCL, diffuse large B-cell lymphoma; KPS, Karnofsky performance score. * 25-/75-percentile range.

**Table 3 cancers-13-02934-t003:** Comparison of survival in non-study vs. study patients

Model	Variable	Hazard ratio	*p*	Lower 95% CI	Upper 95% CI
27 vs. 526 patients, univariate	Arm ^a^	0.93	0.766	0.586	1.483
27 vs. 526 patients, multivariate	Arm ^a^ Age ^b^	0.48 1.39	0.005 <0.001	0.29 1.24	0.80 1.55
	Karnofsky ^c^	0.87	<0.001	0.82	0.92

^a^ Study arm was coded as 1 (study) or 2 (non-study). Hazard ratios > 1 were in favor of study patients, and hazard ratios < 1 were in favor of non-study patients; ^b^ the hazard ratio for age (years) to an age difference of 10 years; ^c^ the hazard ratio for the Karnofsky performance status refers to a change of 10 points.

## Data Availability

The data that support the findings of this study are available from the corresponding author upon reasonable request.

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
