# Peer review of "Patients with Primary Central Nervous System Lymphoma Not Eligible for Clinical Trials: Prognostic Factors, Treatment and Outcome"

_cancers, 2021, doi:10.3390/cancers13122934_

Round 1
Reviewer 1 Report
good work. no more comments
Reviewer 2 Report
The authors have adequately addressed the queries in the revised manuscript. I support the publication of this manuscript.
Reviewer 3 Report
Accept in present form.
This manuscript is a resubmission of an earlier submission. The following is a list of the peer review reports and author responses from that submission.
Round 1
Reviewer 1 Report
This manuscript is showing interesting observation from patient population who can not meet clinical trial eligibility from real world experience. However, no new novel finding is addressed. For example, young age is a good prognostic factor always. Also good response to planned treatment, feasibility to planned treatment should be quite expectable prognostic factors.
Another issue is too small sample size comparing with control arm.
Author Response
Dear Sirs and Madams,
we appreciate the opportunity to resubmit a revised version of our manuscript “Patients with primary central nervous system lymphoma not eligible for clinical trials: Prognostic factors, treatment and outcome” (cancers-1170152) for reconsideration for publication in Cancers as part of the special issue on "Primary CNS Lymphomas: Diagnosis and Treatment”. We would like to thank for the helpful comments of all reviewers, which in our view helped us to improve our manuscript substantially.
We have carefully addressed all reviewers` comments and responded to those on a point-by-point basis. Respective changes in the manuscript are highlighted in yellow.
Reviewer 1:
This manuscript is showing interesting observation from patient population who can not meet clinical trial eligibility from real world experience.
However, no new novel finding is addressed. For example, young age is a good prognostic factor always. Also good response to planned treatment, feasibility to planned treatment should be quite expectable prognostic factors.
REPLY: We are grateful for this reviewer`s comment that the manuscript reports on an interesting observation. While we perfectly agree with the reviewer`s opinion, that no novel prognostic factors have been identified in our series, we feel that in this particular patient population (real life patients not eligible for clinical studies) analyses with regard to prognostic factors are scarce. Yet, the finding that established prognostic factors also apply to patients not eligible for clinical trials is confirmatory and in our view of clinical relevance.
Another issue is too small sample size comparing with control arm.
Response: We completely agree with this reviewer that the number of patients n=34/27 is limited. However, we focused exclusively on patients not eligible for a clinical trial. Such an analysis can hardly be realized within a multicentric setting. Despite a substantial difference in sample size comparing this series with n=411 well documented patients from a clinical trial, we aimed to carry out an analysis on two rather comparable patient cohorts. We addressed this point in the discussion (page 8, lines 255-257).
We hope, that the manuscript in its current version meets the criteria of the editors and the expectations of all reviewers and would like to thank the Reviewers for their valuable comments.
We would like to thank you in advance for devoting your time and attention to our revised manuscript and look forward to hearing from you in due course.
Yours sincerely, on behalf of all authors,
Uwe Schlegel, M.D. Sabine Seidel, M.D. (Corresponding author)
Professor of Neurology Department of Neurology
Head of the Department

Reviewer 2 Report
In this study titled “Patients with Primary Central Nervous System Lymphoma Not Eligible for Clinical Trials: Prognostic Factors, Treatment and Outcome” authors have retrospectively analyzed 34 consecutive patients diagnosed with primary CNS lymphoma who were deemed ineligible for clinical trials during a period of 14 years at their center. 27 patients out of this group were able to receive high-dose methotrexate as treatment and were compared with a group of 411 patients enrolled in G-PCNSL-SG-1 study.
I have the following concerns with this study:
- KPS may be misleading and had inherent limitation of being “spuriously low” in tumors at eloquent locations. How do the authors take this effect into account? Comment.
- One patient was lost to follow up in this study. Did he receive HD-MTX or not?
- H/o prior malignancy or severe heart disease may itself affect mortality significantly. The number of patients in this study is too small to evaluate for this correlation. Larger studies are required to confirm their findings.
- I would also suggest the authors to refer to another similar study which was published in 2014 "Hart A, Baars JW, Kersten MJ, Brandsma D, van Tinteren H, de Jong D, Spiering M, Dewit L, Boogerd W. Outcome of patients with primary central nervous system lymphoma treated outside clinical trials. Neth J Med. 2014 May;72(4):218-23. PMID: 24829178."
Author Response
Dear Sirs and Madams,
we appreciate the opportunity to resubmit a revised version of our manuscript “Patients with primary central nervous system lymphoma not eligible for clinical trials: Prognostic factors, treatment and outcome” (cancers-1170152) for reconsideration for publication in Cancers as part of the special issue on "Primary CNS Lymphomas: Diagnosis and Treatment”. We would like to thank for the helpful comments of all reviewers, which in our view helped us to improve our manuscript substantially.
We have carefully addressed all reviewers` comments and responded to those on a point-by-point basis. Respective changes in the manuscript are highlighted in yellow.
Reviewer 2:
In this study titled “Patients with Primary Central Nervous System Lymphoma Not Eligible for Clinical Trials: Prognostic Factors, Treatment and Outcome” authors have retrospectively analyzed 34 consecutive patients diagnosed with primary CNS lymphoma who were deemed ineligible for clinical trials during a period of 14 years at their center. 27 patients out of this group were able to receive high-dose methotrexate as treatment and were compared with a group of 411 patients enrolled in G-PCNSL-SG-1 study.
I have the following concerns with this study:
- KPS may be misleading and had inherent limitation of being “spuriously low” in tumors at eloquent locations. How do the authors take this effect into account? Comment.
REPLY: We agree with this reviewer that KPS may be misleading, since small lesions in “eloquent” areas of the brain may cause significant impairment, while larger lesions located in other regions may not. However, this limitation applies to all analyses on brain tumors. Furthermore KPS is an established prognostic factor in PCNSL and we would therefore ask for permission not to differentiate between lesions of different location and multifocal vs. unifocal disease. On the other hand, involvement of deep brain structures has been suggested as a prognostic factor in PCNSL, is related to location and has been addressed in the present study (page 5, line 186-188).
- One patient was lost to follow up in this study. Did he receive HD-MTX or not?
REPLY: We thank the reviewer for this comment and included the corresponding information (page 4, line 142+143).
- H/o prior malignancy or severe heart disease may itself affect mortality significantly. The number of patients in this study is too small to evaluate for this correlation. Larger studies are required to confirm their findings.
REPLY: We thank the reviewer for this helpful advice and have addressed this in the manuscript (page 8, line 243-249).
- I would also suggest the authors to refer to another similar study which was published in 2014 "Hart A, Baars JW, Kersten MJ, Brandsma D, van Tinteren H, de Jong D, Spiering M, Dewit L, Boogerd W. Outcome of patients with primary central nervous system lymphoma treated outside clinical trials. Neth J Med. 2014 May;72(4):218-23. PMID: 24829178."
REPLY: We thank the reviewer for this suggestion. The reference is now discussed and included in the manuscript (page 7, line 228+229 and page 8, line 230-235). However, the patient cohort in the study on 52 patients published by Hart et al. is different from the patient population analyzed here, as we included only patients with at least one exclusion criterion for a clinical trial. The study published by Hart et al. included many patients treated outside clinical trials because the histological diagnosis was not confirmed (n=14) or because of a lack of ongoing clinical trials (n=14).
We hope, that the manuscript in its current version meets the criteria of the editors and the expectations of all reviewers and would like to thank the Reviewers for their valuable comments.
We would like to thank you in advance for devoting your time and attention to our revised manuscript and look forward to hearing from you in due course.
Yours sincerely, on behalf of all authors,
Uwe Schlegel, M.D. Sabine Seidel, M.D. (Corresponding author)
Professor of Neurology Department of Neurology
Head of the Department
Reviewer 3 Report
Seidel et al report on retrospective analysis of a small cohort of patients with primary CNS lymphoma treated not on a clinical protocol. This is an important topic and in need of study. Sarid et al, PMID 33099747 is a recent example of such an analysis, and should probably be mentioned in context. The key feature to the analysis by Seidel et al. is comparison to a previously published RCT. While cross-trial comparisons are generally limited, this one is, unfortunately, especially fraught. To start with, the authors compare their treatment-naïve cohort to the n = 411 "ITT" cohort that was randomized only after completing upfront HDMTX (and excluded 140 patients, 66 of whom died from the upfront therapy). This issue should be addressed, and may in part explain the authors' conflicting discussion/conclusion that a) "in non-study patients obviously, treatment results are worse than in study patients" but b) "the multicenter design of the G-PCNSL-SG-1 trial by itself may represent a negative prognostic factor."
Some additional recommendations:
- Account for era of treatment in multivariate analyses, noting that G-PCNSL-SG-1 enrolled from 2000 – 2006, and several advances in managing PCNSL have emerged subsequently that hopefully have translated to improvement in OS.
- What fraction of consecutive patients treated at this Center did versus did not receive treatment on a clinical study..
- P1, line 34: is this median OS?
Author Response
Dear Sirs and Madams,
we appreciate the opportunity to resubmit a revised version of our manuscript “Patients with primary central nervous system lymphoma not eligible for clinical trials: Prognostic factors, treatment and outcome” (cancers-1170152) for reconsideration for publication in Cancers as part of the special issue on "Primary CNS Lymphomas: Diagnosis and Treatment”. We would like to thank for the helpful comments of all reviewers, which in our view helped us to improve our manuscript substantially.
We have carefully addressed all reviewers` comments and responded to those on a point-by-point basis. Respective changes in the manuscript are highlighted in yellow.
Reviewer 3:
Seidel et al report on retrospective analysis of a small cohort of patients with primary CNS lymphoma treated not on a clinical protocol. This is an important topic and in need of study.
Sarid et al, PMID 33099747 is a recent example of such an analysis, and should probably be mentioned in context.
REPLY: We appreciate this reviewer´s statement that the topic of this study is an important one. We have added and discussed the analysis by Sarid et. al in the manuscript (page 7, line 228+229 and page 8, line 230-235). However, the patient population described in the study published by Sarid et al. differs from our patient population, since Sarid et al. included 73 consecutive patients treated with HD-MTX-based regimen at their center. Of these patients, many would have fulfilled inclusion criteria for a clinical trial.
The key feature to the analysis by Seidel et al. is comparison to a previously published RCT. While cross-trial comparisons are generally limited, this one is, unfortunately, especially fraught. To start with, the authors compare their treatment-naïve cohort to the n = 411 "ITT" cohort that was randomized only after completing upfront HDMTX (and excluded 140 patients, 66 of whom died from the upfront therapy).
REPLY: We have carried out the respective comparison, while we acknowledge that any such comparison is limited. We further agree with the reviewer that there are methodological problems within the G-PCNSL-SG1 trial. However, we aimed to find the most appropriate patient group to compare with our cohort. The common clinical and therapeutic feature that applies to both groups, n=27/34 patients not eligible for a clinical trial and n=411/551 eligible for a clinical trial is, that both cohorts had been treated with HD-MTX. One might well imagine other possible side by side comparisons, however, we decided for those two patient populations, since HD-MTX is the most powerful therapeutic factor influencing prognosis. We have addressed this point in the discussion (page 8, lines 264-267).
This issue should be addressed, and may in part explain the authors' conflicting discussion/conclusion that a) "in non-study patients obviously, treatment results are worse than in study patients" but b) "the multicenter design of the G-PCNSL-SG-1 trial by itself may represent a negative prognostic factor."
REPLY: Obviously our statement that results in non-study patients are inferior was misleading and we thank the reviewer for this advice. However, this inferiority this was only due to the lack of MTX application in 7/34 patients. We have changed the manuscript accordingly (page 8, lines 268-273). Furthermore, it is a matter of discussion that a multicentric setting of trials may pose patients at risk for worse treatment outcome in less experienced centers (page 8, 273-276).
Some additional recommendations:
Account for era of treatment in multivariate analyses, noting that G-PCNSL-SG-1 enrolled from 2000 – 2006, and several advances in managing PCNSL have emerged subsequently that hopefully have translated to improvement in OS.
REPLY: We agree with the reviewer and acknowledge that the era of enrollment of 2000-2009 in the G-PCNSL-SG1-study versus 2005-2020 for our patient population is a prognostic factor in favor of the non-study patient population from our center. We have addressed this in the manuscript (page 8, lines 276-283). However, a multivariate analysis regarding era of treatment was not possible from a statistical point of view, since only 5 patients in our study had been diagnosed between 2005-2009 (3 patients with complete remission after treatment, one with progressive disease and one patient with treatment related death).
What fraction of consecutive patients treated at this Center did versus did not receive treatment on a clinical study.
REPLY: We have addressed this point raised by the reviewer and have added the corresponding data in the results section of the manuscript (page 4, lines 135-140).
P1, line 34: is this median OS?
REPLY: We modified the manuscript accordingly (page 1, line 34)
We hope, that the manuscript in its current version meets the criteria of the editors and the expectations of all reviewers and would like to thank the Reviewers for their valuable comments.
We would like to thank you in advance for devoting your time and attention to our revised manuscript and look forward to hearing from you in due course.
Yours sincerely, on behalf of all authors,
Uwe Schlegel, M.D. Sabine Seidel, M.D. (Corresponding author)
Professor of Neurology Department of Neurology
Head of the Department
Round 2
Reviewer 1 Report
well responded. no more comments
Reviewer 3 Report
I cannot endorse the comparison of the 27 patients treated with HDMTX off study and examined retrospectively with the ITT group in G-PCNSL-SG1. The ITT group of G-PCNSL-SG1 excluded many patients that died or were lost to follow-up during treatment with upfront HDMTX. The authors, in response to this issue, state "The common clinical and therapeutic feature that applies to both groups, n=27/34 patients not eligible for a clinical trial and n=411/551 eligible for a clinical trial is, that both cohorts had been treated with HD-MTX." While technically true, this statement ignores the fact that many of the original 551 patients excluded from the 411 ITT group did receive upfront treatment with HDMTX. The outcomes of many of these excluded patients, including those that suffered TRM, would be as comparable to the outcomes of the studied 27 patients as the 411 ITT group's. Indeed, the shape of the KM curves in Figure 3 of the manuscript suggest that about 30% of patients in of the 27 die in the first few months, which seems to be far more in this time period than in the comparison group. In my opinion this needs to be addressed as it is a fundamental message of this manuscript.